# Challenges and enablers in measles vaccination implementation in Ethiopia: Insights from a qualitative study

Gulilat Gezahegn Wodajo[1¤a,¤b*], Tezera Moshago Berheto[2], Haimanot Kifle Telila[3], Yohannes Kebede Lemu[4]

1 Immunization Service Desk, Federal Ministry of Health Ethiopia, Addis Ababa, Ethiopia, 2 National Data Management Center, Ethiopian Public Health Institute, Addis Ababa, Ethiopia, 3 Department of Obstetrics and Gynecology, Zewditu Memorial Hospital, Addis Ababa, Ethiopia, 4 Department of Health, Behavior and Society, Jimma University, Jimma, Ethiopia

¤a Immunization Service Desk, Federal Ministry of Health Ethiopia, Addis Ababa, Ethiopia
¤b Department of Public Health, Texila American University, Georgetown, Guyana
* gimmylv@gmail.com

## Abstract

Even though safe and free vaccines are available, measles vaccination coverage remains low in Ethiopia. There is a paucity of studies on measles vaccination implementation challenges and enablers. Hence, this study aimed to examine implementation challenges and enablers of measles vaccination, both from the perspectives of service providers and caregivers. A case study was conducted in the east Gurage zone of central Ethiopia from December 2023 to May 2024. Fifteen health workers and 16 mothers of children aged 12–23 months who missed the measles vaccination were interviewed. Fifteen service exit interviews and six focus group discussions were also conducted. We analyzed the data manually using an inductive thematic analysis approach and presented the themes and sub-themes with representative quotations. The four basic trustworthiness measures for qualitative research were taken into account. The major measles vaccination implementation challenges for caregivers were inaccessibility, unaware of the next vaccination schedule, unavailability of a daily immunization service, interrupted outreach vaccination sessions, long waiting times, adverse event following immunization, and health workers' impoliteness. Lack of funding, transportation, vaccine refrigerators, space, training, benefit packages, inadequate child screening practices, vaccine stockouts, and fear of vaccine wastage and contraindications were among the major challenges health workers faced when implementing the measles vaccination. Our results suggest that there is an urgent need to improve service availability and accessibility, vaccine and other supply management, basic and refresher training, health workers' benefit packages, and provider-client communications.

**Data availability statement:** All relevant data are within the paper and its Supporting Information files.

**Funding:** The author(s) received no specific funding for this work.

**Competing interests:** The authors have declared that no competing interests exist.

## Introduction

Measles is a highly contagious viral disease that can lead to severe complications, including pneumonia, brain damage, and death [1]. An estimated 136,000 measles deaths occurred worldwide in 2022, with children under the age of five who were either under vaccinated or unvaccinated suffering from the disease [2]. In Ethiopia, where the disease is endemic, cases are reported annually. The yearly count of confirmed cases of measles has dramatically grown since 2021. The number of confirmed measles cases in 2022 was approximately five times higher than that of 2021, and a total of 16,814 laboratory-confirmed measles cases and 182 deaths nationwide occurred between August 12, 2021, and May 1, 2023 [3]. The measles vaccination offers the best protection against the disease. The effectiveness of the measles vaccine is approximately 97% when given in two doses and 93% when given in one [4]. Globally, 83% of children received their first dose of the measles vaccination in 2023, a slight decrease from the 86% coverage in 2019 [5]. Measles containing vaccine first dose (MCV1) and measles containing vaccine second dose (MCV2) coverage in Ethiopia was projected to be 61% and 53%, respectively, in 2023, according to WHO UNICEF Estimates of National Immunization Coverage (WUENIC) [6].

Implementing an immunization program involves several of components: service delivery strategies, vaccine supply, cold chain storage, logistics, vaccine safety, human resources, training, data use, communication, and financing. The effectiveness of vaccination programs relies on the challenges and enablers of these program implementation components [7]. There are studies conducted in Ethiopia with an emphasis on the barriers that prevent children from getting the measles vaccine at the individual, community, and, to a lesser degree, service delivery levels [8–14]. The individual-level barriers related to socio-demographic characteristics identified in these studies included maternal educational level, maternal age, marital status, geographical location of residence, and being in the birth order of 1−3 [9,11]. Awareness and access-related barriers were: lack of knowledge and awareness, poor attitude and satisfaction toward child vaccination, forgetting the next vaccination schedule, migration of parents from one place to another, work overload, insufficient resources, not participating in the women's development army, rumors and misinformation, and inconvenient road conditions [10–12]. Other barriers described in these quantitative studies include limited household visits by health extension workers, closed health posts, restricted availability of immunization services, vaccine vials not opened for a few children, vaccine stockouts, shortages of cold chain technologies, data inaccuracy, and data incompleteness [9,10,12–14]. However, there is a paucity of qualitative studies that have investigated in depth the challenges and enablers of measles vaccination implementation in Ethiopia.

Hence, this qualitative study aimed to examine implementation challenges and enablers to the uptake of the first and second doses of measles vaccination, both from the perspectives of service providers and caregivers.

## Conceptual framework

The conceptual framework used in this research is adapted from the WHO Health System Building Blocks framework [15]. It shows the measles vaccination implementation challenges and enablers in Ethiopia (Fig 1).

## Materials and methods

### Ethics statement

We obtained an ethical approval certificate by the Institutional Review Board, Ethiopian Public Health Institute (IRB-EPHI). A permission letter was also received from the local government (Sodo and South Sodo district health offices) to undertake the study. After explaining the purpose of the study, privacy, and confidentiality issues, written informed consent was obtained from each study participant. For caregivers with low literacy, we explained the consent form verbally to them in the presence of a literate family member or neighbor before they agreed to participate. During a focus group discussion, we invited each participant for their written consent to participate in the study. Everyone in the group asked to agree to keep what's discussed confidential and to respect each other's privacy. We disclosed all possible risks of harm to participants before the study to get informed consent. The study subjects' right to refuse was respected. Identification of study participants by name was also avoided to assure the confidentiality of the information obtained

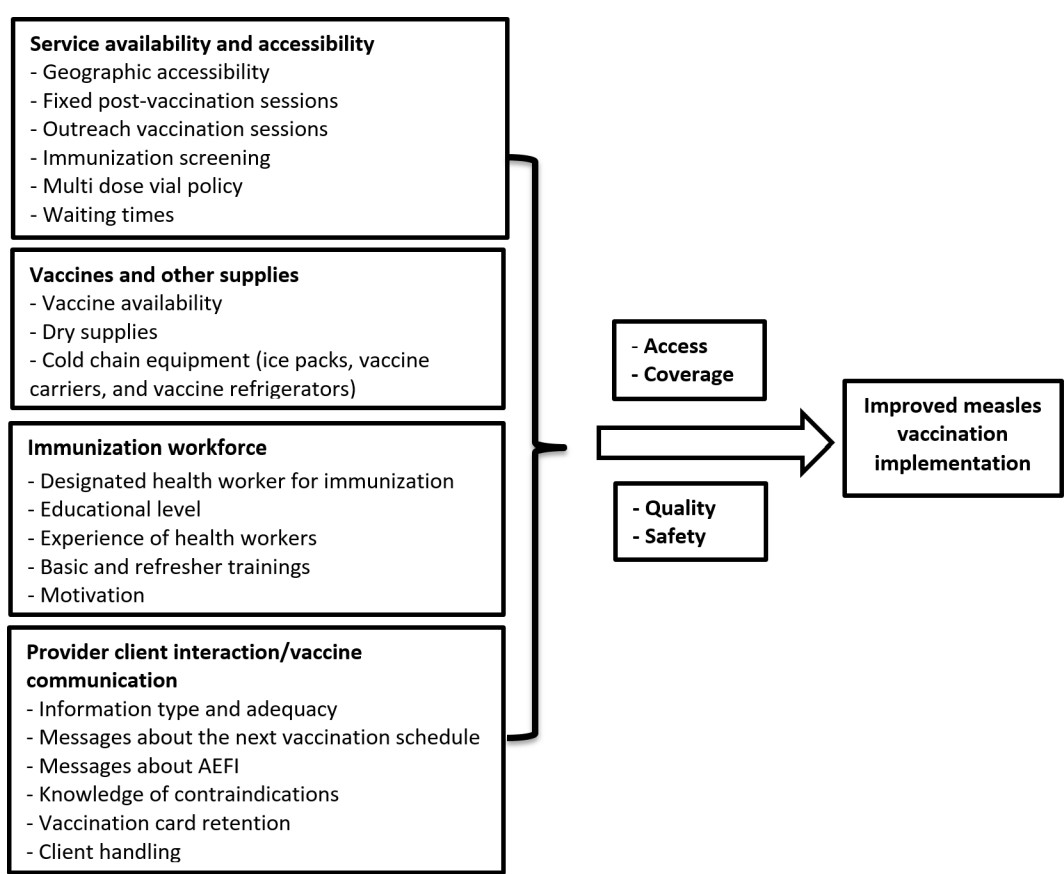

**Fig 1. A conceptual framework that shows measles vaccination implementation challenges and enablers in Ethiopia.**

## Study design

This study employed a case study research design to comprehensively investigate measles immunization service delivery implementation challenges and enablers.

## Study setting

The study was conducted across purposively selected health posts, health centers, and district health offices in the east Gurage zone of central Ethiopia from December 2023 to May 2024. Sodo and South Sodo districts were selected because of the high number of unvaccinated children in the area who needed to receive the measles vaccine [16]. There were 33 private clinics, 103 health posts, 71 health centers, and two hospitals in the zone. According to the population projection from the 2007 census, the total population and surviving infants in the study area were estimated to be 502,404 and 16,000, respectively [17].

## Participants and sampling

A total of fifteen health workers from the low-performing districts and health facilities (02 district health office heads, 02 district Maternal and Child Health (MCH) coordinators, 03 Health Center (HC) heads, 03 HC Routine Immunization (RI) focal persons, and 05 Health Post (HP) heads/Health Extension Workers (HEWs)) were selected purposively for key informant interviews. Fifteen mothers with children aged 12–23 months who sought any services at a health facility were also randomly selected for a health facility service exit interview. A total of 16 mothers of children aged 12–23 months in low-performing kebeles who missed the first or second dose of measles vaccination were selected purposively for in-depth interviews. Six focus group discussions (FGDs) were conducted in selected low-performing kebeles (Fig 2).

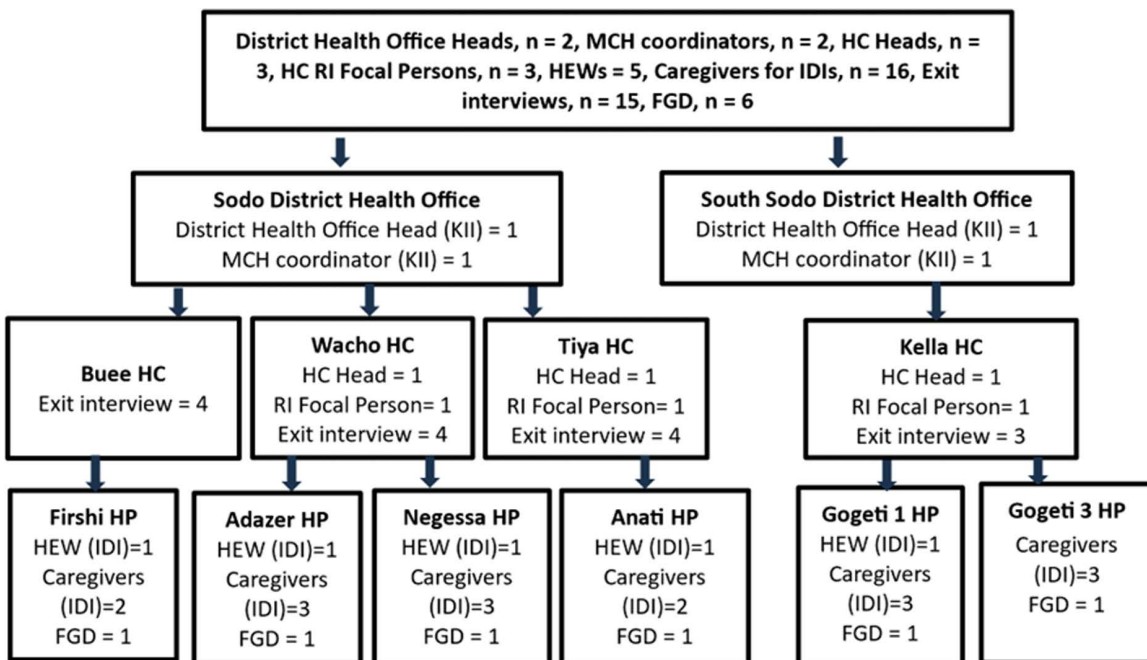

**Fig 2. Schematic illustration of the sampling procedure.**

## Measurement

Participant health workers were asked to answer open-ended questions regarding perceived and experienced measles vaccination implementation challenges and enablers, whereas mothers were asked to explain the reasons why their children missed measles vaccination through an in-depth interview (IDI). Conversely, through FGD, women over 50 and women aged 15–49 with children under five were asked about their experiences and perceived reasons for children missing or dropping measles vaccination. During the exit interview, mothers were asked about their experience of care during the vaccination session.

## Data collection methods

Data was collected through an in-depth interview, key informant interview (KII), service user exit interviews, and focus group discussions. The in-depth interview, key informant interview, exit interview, and FGD guides were adapted from WHO [18,19]. These data collection tools were translated into Amharic and reviewed by bilingual experts to guarantee translation correctness and consistency. Three public health specialists with master's degrees who have experience in collecting qualitative data were recruited. A two-day training was provided before deploying them to the field. The guides were pretested in a similar setting outside of the study area. Questions and translations were refined before data collection based on feedback from the pretest. Data were collected face-to-face from health workers at work places, from caregivers at their homes, and from FGD at their village. The duration of the interviews varied from 40 to 90 minutes, depending on the saturation of the participants' ideas, with an average duration of 50 minutes.. Note-taking and tape recorders were also employed to collect the data.

## Data analysis

Using an inductive thematic analysis approach, data were manually coded, categorized, and analyzed. The interviews were done in Amharic, and the data were first transcribed in that language. The information was then translated into English. Experienced translators who are fluent in both English and Amharic and who have a deep understanding of the vaccination program were employed for the translation. Back translation, proofreading, and editing were carried out to ensure the accuracy of the translation. Initially, in order to get a general picture of the data, we read and reread the audio-taped transcripts. Two investigators independently reviewed the transcripts to pinpoint important themes and create a code structure, one holding a PhD degree and the other a master's level. Then, a final edition of the code was developed, and the categories and themes were constructed. We proceeded with additional categorization and data reduction to identify sub-themes. Finally, we presented the themes and sub-themes with representative quotations.

## Trustworthiness

The four basic trustworthiness measures for qualitative research were taken into account during the whole process [20]. The first criterion was credibility. It was attained by making sure the interviewers had the experience and research skills to carry out their roles. Interviewers spent an average of three days per kebele or health post interacting with caregivers and one day at the health center with health workers. Interview tools were tested at two induction meetings and using three pilot interviews. Interviewers submitted all field notes for analysis and storage, and we created a comprehensive record of the data collection process. Moreover, we combined different data collection techniques (in-depth interviews, KIIs, exit interviews, and FGD) and quantified operational and theoretical data saturation.

The second criterion was dependability, which was attained through documenting each step of the research process and decisions made, including changes in the data collection process, translation, and analysis. The third criterion, transferability, was addressed through describing the research context, the sampling methods, participants, and the criteria for participant selection to assist in determining whether the findings might be transferable to similar populations or settings

outside the study area. The fourth criterion was confirmability, which was attained through peer debriefing to minimize personal biases and validate findings. Few key informants participated in the verification process to ensure their viewpoints were accurately captured (Table 1).

## Results

### Socio-demographic characteristics of participants

A total of fifteen health workers from the low-performing districts and health facilities (02 district health office heads, 02 district MCH coordinators, 03 HC heads, 03 HC RI focal persons, and 05 HP heads/HEWs) participated as key informants. Eight of the participant health workers were male, and the remaining seven were female. Their ages ranged from 20 to 49.

**Table 1. Socio-demographic characteristics of caregivers.**

| Variables | | IDI (caregivers) | | FGD (caregivers) | | Exit interview (caregivers) | |
|---|---|---|---|---|---|---|---|
| | | # | % | # | % | # | % |
| Age of children | 12–23 months | 16 | 100 | NA | NA | NA | NA |
| Age of mothers/ health workers | <20 | 0 | 0 | 2 | 3 | 0 | 0 |
| | 20–29 | 4 | 25 | 6 | 8 | 4 | 27 |
| | 30–39 | 9 | 56 | 52 | 72 | 8 | 53 |
| | 40-49 | 3 | 19 | 9 | 13 | 3 | 20 |
| | ≥50 | 0 | 0 | 3 | 4 | 0 | 0 |
| | Total | 16 | 100 | 72 | 100 | 15 | 100 |
| Sex | Male | 0 | 0 | 0 | 0 | 0 | 0 |
| | Female | 16 | 100 | 72 | 100 | 15 | 100 |
| | Total | 16 | 100 | 72 | 100 | 15 | 100 |
| Marital status | Married | 14 | 88 | 67 | 93 | 15 | 100 |
| | Divorced | 1 | 6 | 2 | 3 | 0 | 0 |
| | Widow | 1 | 6 | 3 | 4 | 0 | 0 |
| | Single | 0 | 0 | 0 | 0 | 0 | 0 |
| | Total | 16 | 100 | 72 | 100 | 15 | 100 |
| Educational status | No formal education | 15 | 94 | 58 | 81 | 12 | 80 |
| | Primary | 1 | 6 | 13 | 18 | 3 | 20 |
| | Secondary | 0 | 0 | 1 | 1 | 0 | 0 |
| | University | 0 | 0 | 0 | 0 | 0 | 0 |
| | Post graduate | 0 | 0 | 0 | 0 | 0 | 0 |
| | Total | 16 | 100 | 72 | 100 | 15 | 100 |
| Residence | Urban | 0 | 0 | 0 | 0 | 4 | 27 |
| | Rural | 16 | 100 | 72 | 100 | 11 | 73 |
| | Total | 16 | 100 | 72 | 100 | 15 | 100 |

### Key themes

In this study, we identified four themes and seventeen sub-themes of perceived and experienced measles vaccination implementation challenges and enablers. These are presented as follows:

**Theme: Service availability and accessibility**

This theme contains factors related to geographic inaccessibility, the unavailability of outreach vaccination sessions, fixed post-vaccination session interruptions, poor child screening practices, fear of vaccine wastage, and long waiting times.

**Geographic inaccessibility**

Evidence from caregivers and health extension workers showed that inaccessibility due to distance, lack of transport, mountainous terrains, and security threats were reported as major challenges for measles vaccination service. A 29-year-old mother with a child who was a defaulter said:

> "*My village is far from the health post, and there is no transport access; it took me more than four hours to make a round trip by foot. During the rainy season, we miss vaccination sessions because of overflowing rivers.*" One of the HEWs also said:

> "*Some villages, like "Semero," take five hours for a round trip by foot.*"

**Unavailability of outreach vaccination sessions**

Our analysis revealed that there had been vaccination services at a house-to-house level and outreach vaccination posts. However, recently, the outreach immunization programs have either stopped being offered or have been severely disrupted. A 53-year-old female FGD participant said:

> "*There used to be an outreach program and a house-to-house vaccination service in our community. This time around, however, it is only available at the health post level, which is not convenient for mothers coming from a distance.*"

The provision of vaccination services in border areas that are difficult to reach was hampered by a lack of funding for transportation and security concerns. A district health office head reported:

> "*We don't have funds for transportation to provide outreach services in hard-to-reach communities. Additionally, we have been facing security challenges recently. However, those remote areas are prone to measles and other vaccine-preventable disease outbreaks.*" This idea was supported by one of the HEWs:

> "*Due to workload and financial constraints, we were unable to implement the outreach and house-to-house vaccination services.*"

**Fixed post vaccination session interruption**

Health facilities are expected to provide immunization services on a daily basis. However, fixed post vaccination session interruptions were reported both at the health centers and health posts. A health extension worker from the health post said:

> "*Due to the fact that I was on maternity leave, this health post was closed for an extended period of time.*" This idea was supported by a 28-year-old mother with a child who missed the measles vaccination:

> "*I went to the health post, but it was closed when I arrived. I visited the health post once more the next month, but it was closed. Considering that the health post is often closed, I decided not to go for the immunization service after three rounds of attempts.*" One health center RI focal person also reported:

> "*Due to a shortage of space, we are unable to offer vaccination services on a daily basis.*"

In this study, the unavailability of a daily immunization service at health facilities was identified as one of the causes of missed opportunities. A caregiver with a child who missed the measles vaccination at the service exit interview:

*"I came to this HC for medical consultation for my sick child, and he screened my child's immunization status and told me that my child is eligible for the measles second dose; however, he didn't refer me to the vaccination room because today is not a vaccination date in this facility."*

### Poor child screening practices

Another identified reason for missed measles vaccination was the absence of screening practices for children attending health institutions for purposes other than vaccinations. A mother with a measles vaccine-eligible child at the exit interview said:

*"I came to this health center with someone. I didn't know that my baby was eligible for vaccination; even the health worker who saw us did not tell me about my child's vaccination status."* This idea was supported by one of the HC RI focal persons:

*"We only screen children at under five clinics because the chart booklet obligates us to screen the immunization status of the child. However, mothers can come with their children for family planning or other services, and we usually don't screen children's immunization status in these departments."*

### Fear of vaccine wastage

Measles vaccine doses per vial presentation were also mentioned as the main cause of low measles vaccination performance in the study area. The multi-dose vial policy states that the measles vaccines that have been reconstituted and are not used within six hours should be discarded. A single vial contains ten doses of the measles vaccine, and HEWs have to keep the vaccine wastage rate below 25%. However, the HEWs reported that it is challenging to get up to seven eligible children in one vaccination session to open one vial of the measles vaccine. One HEW said:

*"We couldn't get 10 eligible children to vaccinate against measles in one vaccination session. To minimize vaccine wastage, especially in low-population kebeles, we will wait up to three months to get at least seven eligible children. While we wait to get these children, some of them will be missed or defaulted. Thus, measles vaccines ought to be supplied in vials containing only one or two doses."* The idea was supported by a 31-year-old defaulter mother:

*"We were three in number for the measles' vaccination during the session. The HEW told us to come back next month if she gets an additional four eligible children for measles, and then she will open the vial for us. However, we were still the only eligible children, and she again appointed us for the second time. Finally, I decided not to go back."*

### Long waiting times

Respondent health workers and caregivers mentioned that there were long waiting times during the vaccination sessions. The workload and HEW's duty to collect vaccines from the health center were mentioned as a cause of long waiting times at the health post level during a vaccination session. One HEW said:

*"I usually collect vaccines from the nearby health center during a vaccination session; the mothers may find this inconvenient because of the long waiting times."* This idea was supported by a 38-year-old mother with a child who was a defaulter:

*"We usually face long waiting times during the vaccination sessions; it took almost half a day, including the round trips. There should be shorter waiting times because we are farmers."*

## Theme: Immunization workforce

This theme embraces the educational level of the HEWs, experiences of the HEWs, training gaps, and low HWs motivation.

### Educational level of the HEWs

It's surprising to learn that health extension workers who have upgraded to Level 4 in their education were not dedicated to offering community-based immunizations and other preventive health services. One of the HC heads reported:

*"Health extension workers' commitment usually decreases when they return from level 4 upgrading. They are not visiting households as they did previously. Upon upgrading to level 4, I think they expect more curative services than preventive and home visits."*

### Experience of the health extension workers (HEWs)

Our analysis revealed that recently hired young HEWs performed better in immunization services than the older or senior HEWs. One of the district health office heads reported:

*"All HEWs are females, and I believe that senior health extension workers are getting tired of providing routine health-care in rural areas because of their increased age and other maternal factors, which have led to low immunization performance. However, recently hired HEWs are youthful and sufficiently active to offer community-level services in the hard-to-reach communities with bad terrain."*

### Training gaps

Respondent health workers mentioned a shortage of training as one of the constraints in the immunization program. Unlike in previous years, health extension workers didn't receive basic and refresher training from the Ministry of Health and other implementing partners. A district health office MCH coordinator reported:

*"When new health extension workers are deployed to the system, they should get basic EPI training. However, this is not in place, and during new vaccine introductions, we face challenges due to training gaps."* One HC head also said:

*"We have only one trained nurse for this service at the health center level. It will be difficult to deliver quality services in the absence of trained health workers."*

### Low HEWs motivation

Our analysis revealed that HEW's commitment to trace and vaccinate defaulter children has decreased. One of the HC RI focal persons reported:

*"Similar to past years, the HEWs require an incentive structure. The amount of work has increased recently, and health extension workers are not being motivated by any training, rewards, or other benefit plans. If a mother missed one vaccination session due to a social problem, some health extension workers are reluctant to find and vaccinate the child in the next session."* This idea was supported by one of the HEWs:

*"We don't have benefits packages like rewards or incentives. However, the volume of work is increasing day by day. The health office should motivate the HEWs to improve healthcare services at the community level."*

### Theme: Vaccines and other logistics shortages

**Vaccine stockout.** Occasional vaccine stockouts were reported in the study area. Health facilities usually request vaccines based on their needs. However, the supplier refills based on their stock on hand. As a result, there were occasional stockouts of measles vaccines in the study area. Respondent caregivers also witnessed the presence of vaccine stockouts at the health facility level. A 27-year-old FGD participant said:

*"I think there is a shortage of vaccines in this health post, especially the measles vaccine. They usually gave us a long appointment to get our children vaccinated for measles."*

### Lack of vaccine refrigerator

Vaccine refrigerators play a crucial role in helping to keep vaccines at the proper temperature, which is essential for their efficacy; if vaccines are exposed to extreme temperatures, either too hot or too cold, they can lose their ability to provide immunity against diseases, rendering them ineffective and potentially compromising the entire immunization program. The unavailability of vaccine refrigerators at the health post level was found to be a barrier to measles vaccination in the study area. A district health office head said:

*"Health posts are not providing immunization services on a daily basis because of a lack of vaccine refrigerators. Which has compromised our immunization performance significantly. It would be great if our health posts could have solar refrigerators."*

### Theme: Provider-client interaction

This theme embraces information type and adequacy, messages about the next vaccination schedule, messages about Adverse Event Following Immunization (AEFI), fear of contraindication, and health workers' impoliteness.

### Information type and adequacy of messages

Key messages during vaccination sessions focus on the importance and effectiveness of vaccines, explaining the vaccination schedule, addressing common concerns about vaccines, and reminding parents to bring their children for all scheduled doses on time, while also providing information on potential side effects and how to manage them. However, there were gaps in addressing key messages during the vaccination session in the study area. Out of 15 interviewed caregivers at the service exit, seven didn't receive information about vaccine-related reactions or other key messages after vaccination.

### Messages about the next vaccination schedule

Most of the interviewed mothers at the service exit didn't know the measles second dose vaccination schedule. A mother with an 18-month-old child at the exit interview:

*"I came to this health center for child medication. I thought that my child had completed all the required vaccinations when he took the measles vaccine at nine months. I was unaware of the second dose of measles vaccination, which was at 15 months. I wish the HEW had reminded me."*

Respondent health workers also mentioned that the measles second dose is administered at fifteen months, in the second year of life, and it is a lengthy appointment. As a result, mothers usually forget the schedule.

### Adverse events following immunization (AEFI)

Respondent health workers explained how vaccination against measles presents more difficulties than the others due to adverse events following immunization. One of the HEWs workers mentioned:

*"At 14 weeks, children receive three injections (Penta3, PCV3, and IPV) in addition to two orally administered vaccines. It is common to develop local reactions, fever, and other adverse events following immunization. Therefore, mothers become reluctant to come at 9 months for the first dose of measles unless we counsel them very well."*

Respondent mothers were also mentioned as having insufficient health information about AEFI during immunization sessions. One FGD participant said

*"My first child didn't take the measles vaccination. Because my child became sick after he received three injections at the age of fourteen weeks. My child had a fever, couldn't crawl, and was unable to sleep. I therefore declined the measles vaccination, which was scheduled at 9 months. If I had been aware of adverse events following immunization, I wouldn't have refused."*

### Fear of contraindications

Caregivers and some health workers were hesitant to vaccinate children who were sick or on medication because they were worried about the vaccine's contraindications. A 31-year-old mother with a sick baby at the service exit interview said:

*"I came in today to obtain medical attention for my sick child. I know my child needs to be vaccinated against measles, but the medication may not work well with it. As a result, I decline to vaccinate my child."*

Some health workers also stated that they were hesitant to vaccinate sick children due to concerns about possible contraindications to other treatments. A 36-year-old health worker said:

*"I knew that the child had not had a measles vaccination. However, I didn't offer vaccination service because the child was febrile and taking medication. I wanted to be safe in case something untoward occurs following the vaccine or as a result of contraindications."*

### Health worker's impoliteness

Some respondent caregivers mentioned that one of the reasons for missed measles vaccinations was the impoliteness of health workers. A 33-year-old mother with a measles vaccine-missed child said:

*"Because of a conflict with my husband, I went to my family's village and missed the measles vaccination schedule. I didn't visit the health post when I returned because of fear of the health extension worker."*

A child who loses their vaccination card is likely to miss their vaccinations and lose vital health information. Therefore, replacing lost vaccination cards might minimize the risk of possible defaulters. However, some respondent mothers mentioned that they were afraid of the health extension workers to ask for a replacement. A 34-year-old mother with a child who missed the measles vaccination reported:

*"I misplaced the immunization card, so I missed my child's scheduled vaccinations. I was afraid to ask for a replacement card because the HEW told us to keep it somewhere safe."*

## Discussion

This study aimed to examine implementation challenges and enablers to the uptake of the first and second doses of measles vaccination, both from the perspectives of service providers and caregivers. Our study identified the key experienced

implementation constraints for the measles vaccination. The identified constraints were related to the following four thematic areas: service availability and accessibility, the immunization workforce, vaccine and other logistics shortages, and provider-client interaction related factors.

Geographic inaccessibility was one of the factors that hindered children from receiving the measles vaccination. Mothers living in remote and insecure villages were unable to visit health posts for the nine-month measles vaccination. In addition, outreach vaccination services were highly interrupted and not available in some areas due to a shortage of transportation. This finding is consistent with studies done in Ethiopia [8,10–12] and Nigeria [21]. Re-establishing outreach services might be mandatory to reach every child in those communities [22].

Our analysis revealed that there was no daily immunization service in the study area. Immunization services were only offered once a month by health posts and once a week by health centers. Moreover, other than immunization and under-five clinics, there were no procedures in place for screening children for vaccinations whenever they went to a health facility with their families. This might be one of the reasons for poor measles vaccination performance in the study area. This finding is in agreement with previous studies done in Jimma, Ethiopia [23] and Kenya [24]. However, screening children for vaccination during any contact with health services, vaccinating on the spot, and providing immunization services on a daily basis might improve immunization coverage by reducing missed opportunities for vaccinations. Additionally, restrictions on opening measles vaccine vials with the intent to reduce vaccine wastage also caused another missed opportunity. Vaccinators usually follow the rule that requires them to open a measles vaccine vial only in the presence of seven or more children who are scheduled for measles vaccination. As a result, health extension workers wait up to three months without opening the measles vaccine vial in low-populated kebeles. The finding is in line with previous studies done in Gonder, Ethiopia [25], and Kenya [24]. This practice could increase the probability of dropping or missing the measles vaccination. Switching the existing 10-dose measles vaccine vial to a 5-dose might improve measles vaccination uptake by allowing health workers to open the vial with a small number of eligible children and reduce missed opportunities.

Some defaulter mothers reported that they defaulted because of the long waiting times at health facilities. Mothers had to spend more time at the health posts, up to a half day, because health extension workers were obligated to collect vaccines from the health centers for each vaccination session. This might have discouraged mothers from coming back for the subsequent doses since the majority of them are farmers and have a lot of things to do at home. The finding is in agreement with previous studies done in Ethiopia [9,26,27] and sub-Saharan countries [28]. The issue might be fixed if the health posts could store vaccines or the health center delivered the vaccines to the health post during a vaccination session.

Health posts were closed for an extended period while health extension workers were on maternity leave. Considering that the health post is often closed, mothers decided not to go to the immunization service after several attempts. This finding is consistent with a study done in Ethiopia [14]. Deploying new health extension workers or mobilizing HEWs from the nearby health posts might be a temporary solution to retain children on the immunization schedule in such communities. This study also revealed that the recently hired HEWs are youthful and sufficiently active to offer community-level services in hard-to-reach communities with bad terrain, whereas senior health extension workers were not provided healthcare services sufficiently in rural areas, which has led to low immunization performance. All HEWs in the study area are females, and as they are getting older, a number of maternal-related factors might have contributed to their poor performance in comparison to the newly hired HEWs. Surprisingly, health extension workers who have upgraded to Level 4 in their education were not committed to offering immunization services at the community level when compared to Level 3 HEWs. This might be due to burnout from a prolonged period of difficult work in the community or a desire to provide clinical services at the health center level rather than preventive ones at the community level. This finding is consistent with a study done in Ethiopia [29] and against a study done in Indonesia [30]. The difference might be due to study setting variation because our finding was from a health post or community level, whereas a study in Indonesia was in a hospital setting.

Unlike in previous years, health extension workers didn't receive basic or refresher training, and there was only one trained nurse assigned to provide vaccination services at the health center level. Vaccination sessions were interrupted

whenever the trained health worker was absent for various reasons. This might be contributed to missed measles vaccinations and compromised the quality of the service. Moreover, it seems HEW's commitment to trace and vaccinate defaulter children has decreased. One of the reasons mentioned by the health extension workers was that the workload has increased recently; however, health extension workers were not being motivated by training, rewards, or other benefit packages. This finding is in agreement with studies done in Ethiopia [31] and Greece [32]. One of the most important components of improving the immunization program is training the health professionals [33]. Hence, the provision of basic training, rewards, and other benefits might improve measles vaccination in the study area.

In this study, the unavailability of vaccine refrigerators was found to be a barrier to measles vaccination at the health post level. It was mentioned as the primary cause of health posts' failure to offer daily immunization services. The finding is in line with studies done in Ethiopia [27] and Kenya [34]. Since health posts were constructed in rural communities in the absence of electric power, the main alternatives are kerosene or gas-driven refrigerators and solar refrigerators. Due to their low operational cost, solar refrigerators might be a plausible solution for vaccine storage at the health post level [35]. Measles vaccinations have occasionally been interrupted because of vaccine stockouts at health facilities. This finding is in line with a study done in Ethiopia [14]. Enhancing the supply chain and logistics system could guarantee the continuous provision of high-quality vaccines from the point of manufacture to the service delivery stage and prevent the loss of immunization opportunities due to vaccine shortages [36].

Our analysis revealed that key messages were not shared with caregivers during the immunization sessions. As a result, children were found to have missed measles vaccination due to insufficient vaccine communications. Children receive two oral vaccines and three injections (Penta3, PCV3, and IPV) at 14 weeks. After receiving these vaccines, it is expected that children develop local reactions, fever, and other mild AEFIs. Because of insufficient information about these adverse events following immunization, mothers became reluctant to come back at 9 months for the first dose of the measles vaccine. Moreover, mothers were not willing to vaccinate their children, even with minor illnesses, including low-grade fever, due to a fear of contraindication to other treatments given. The finding is consistent with previous studies done in Ethiopia [12,37] and Zambia [38]. This could be the result of inadequate health information on contraindications.

Most of the interviewed defaulter mothers at the service exit and IDIs didn't know the measles second dose vaccination schedule. This finding is in agreement with studies conducted in Ethiopia [10,12] and Pakistan [39]. A plausible explanation could be that the second dose of measles vaccine is given at fifteen months, during the second year of life, and requires a lengthy appointment. Hence, mothers may overlook the timetable. It may be crucial to remind them of it before the planned session. Our analysis revealed poor provider-client interactions, which affected measles vaccination. Some respondent mothers reported that health workers were disrespecting them and treating them poorly during vaccination sessions. This result is in line with studies done in Ethiopia [12,40] and Pakistan [39]. Child immunization programs might be helped or hindered by the quality of interactions between healthcare providers and parents. Because child immunization requires several service visits, interactions between health workers and caregivers might impact vaccination coverage and dropout rates [41].

## Implication for policy and practice

The fixed post-vaccination sessions were frequently interrupted, and most health posts did not provide daily immunization services due to a lack of vaccine refrigerators. Even those who had vaccine refrigerators were not functional because of the absence of electric power. We suggest that availing solar refrigerators for health posts might be a plausible solution because of their low operational cost. Unavailability of outreach vaccination sessions was also a major finding in this study. It resulted from a lack of funding for transportation to reach hard-to-reach areas. We suggest that allocating funds for outreach vaccination sessions, especially for the hard-to-reach communities, and re-establishing the outreach vaccination program might improve service accessibility. Restrictions on opening measles vaccine vials with the intent to reduce wastage resulted in children missing measles vaccination. We suggest that switching the measles 10-dose vial to

a 5-dose vial might improve measles vaccination uptake and reduce missed opportunities. Moreover, poor child screening procedures in healthcare facilities resulted in missed opportunities to vaccinate eligible children. We propose that each case team conduct an eligible children screening for those who visit the health institution for any other services.

Our study identified that basic immunization training and other benefit and incentive packages were not provided to health extension workers. It may have contributed to the poor performance of the measles vaccination. Therefore, in order to increase the motivation of health workers and the quality of services, we recommend that the Ministry of Health offer basic immunization training, incentives for good work, and other benefit packages. Due to inadequate information provided during vaccination sessions and poor service provider-client interaction, children missed immunization services. We suggest that service providers should improve their provider-client interactions and strictly monitor and address key messages during vaccination sessions, like the type of vaccine the child received, possible AEFI, contraindications, the next vaccination schedule, and keeping the vaccination card in a safe place.

### Strengths and limitations of the study

Because of its qualitative nature, this study has the advantage of better understanding measles vaccination barriers and enablers from both providers and caregivers' perspectives that prior quantitative findings were unable to address. We combined different data collection techniques (in-depth interviews, KIIs, exit interviews, and FGD) with individual and program dimensions. In addition, it identified the clients' perception and described how they felt and experienced those touchpoints. We trained interviewers to be neutral, use open-ended questions, and ensure consistent interview protocols to minimize Interviewer Bias. Confirmability was attained through peer debriefing to minimize personal biases and validate findings. This study has its limitations of a qualitative finding nature. Even though the study samples were determined by the idea saturation of the study participants, they might not be representative of the general and diverse population, and the results cannot be generalized to other settings. In addition, the study encountered limitations, such as limited qualitative literature on vaccination to use in the discussion section for a comparison of the finding with other articles. Instead, some mixed studies were used due to a limited number of qualitative articles on child vaccination.

### Conclusions

The major measles vaccination implementation challenges for caregivers were inaccessibility, unaware of the next vaccination schedule, unavailability of a daily immunization service, interrupted outreach vaccination sessions, long waiting times, AEFI, and health workers' impoliteness. Lack of funding, transportation, vaccine refrigerators, space, training, benefit packages, inadequate child screening practices, vaccine stockouts, and fear of vaccine wastage and contraindications were among the major challenges health workers faced when implementing the measles vaccination. Our results suggest that there is an urgent need to improve service availability and accessibility, vaccine and other supply management, basic and refresher training, health workers' benefit packages, and provider-client communications.

### Supporting information

**S1 Data. Mothers/caregivers in-depth interview guide targeted for assessing reasons for measles vaccination dropout.**
(DOCX)

### Acknowledgments

We would like to thank MOH Ethiopia, Sodo, and South Sodo district health offices for their cooperation. Our appreciation also goes to health workers working at all levels and directly involved in the data collection. Finally, our gratitude also goes to the study participants for providing us with their time and giving us important information.

## Author contributions

**Conceptualization:** Gulilat Gezahegn Wodajo.

**Data curation:** Tezera Moshago Berheto, Haimanot Kifle Telila.

**Formal analysis:** Gulilat Gezahegn Wodajo, Tezera Moshago Berheto, Haimanot Kifle Telila.

**Investigation:** Gulilat Gezahegn Wodajo, Haimanot Kifle Telila.

**Methodology:** Gulilat Gezahegn Wodajo, Yohannes Kebede Lemu.

**Supervision:** Gulilat Gezahegn Wodajo, Yohannes Kebede Lemu.

**Validation:** Gulilat Gezahegn Wodajo, Yohannes Kebede Lemu.

**Visualization:** Gulilat Gezahegn Wodajo.

**Writing – original draft:** Gulilat Gezahegn Wodajo.

**Writing – review & editing:** Gulilat Gezahegn Wodajo, Yohannes Kebede Lemu.

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
