## [Decision Letter · Decision Letter 0]

PGPH-D-24-02245

Challenges and enablers in measles vaccination implementation in Ethiopia: Insights from a qualitative study

Dear Dr. Wodajo,

Thank you for submitting your manuscript to PLOS Global Public Health. After careful consideration, we feel that it has merit but does not fully meet PLOS Global Public Health’s publication criteria as it currently stands. Therefore, we invite you to submit a revised version of the manuscript that addresses the points raised during the review process.

We look forward to receiving your revised manuscript.

Kind regards,

Orvalho Augusto, MD, MPH, PhD

Academic Editor

Journal Requirements:

1. We have amended your Competing Interest statement to comply with journal style. We kindly ask that you double check the statement and let us know if anything is incorrect.

2. In the online submission form, you indicated that The data that support the findings of this study are available from the corresponding author upon reasonable request. 

a. In a public repository, 

b. Within the manuscript itself, or 

c. Uploaded as supplementary information.

Additional Editor Comments (if provided):

This is a relevant report accounting for the challenges of measles immunization somewhere in Ethiopia as well as in many similar settings globally. Here are a few comments/suggestions/questions:

1. Keywords - I suggest adding a few more.

2. Introduction:

- Lines 43 and 44 could be shortened into just a single sentence.

- Lines 47 and 48. Is it a significant decrease from 86% to 83% [in the substantive matter, not just statistically]?

- Paragraph involving lines 60 to 65 speaks of "implementation challenges and enablers". What is this? Please add in the background an introduction of the concept of "implementation challenges and enablers". The whole manuscript depends on this.

- Also, is "service delivery" not part of "implementation"? This is apropos line 60.

- Also, 64 speaks of "quantitative studies" being limited. Please expand this.

3. Please have a figure with a conceptual framework or model.

4. Line 85 - where it says that women were recruited from "exit visits". Please specify what kind of exit visits are.

5. Line 101 introduces the abbreviation KIIs. Please fully state what it means in the first mention.

6. Methods:

- For data collection tools, please add them as supplementary materials.

- Lines 114 and 115 - Was any action taken to guarantee that the translation to English was accurate

- Lines 117 - It is okay that a PhD and master's were used. But this doesn't tell us about their study-relevant experience. Are colleagues specialists in Public Health?

- Line 124 - about the four measures of trustworthiness. Please add a better citation than the current citation 14.

7. Lines 162 to 164 are not for socio-demographics. Please have a title indicating that you transitioned to themes.

Reviewers' comments:

Reviewer's Responses to Questions

**Comments to the Author**

1. Does this manuscript meet PLOS Global Public Health’s publication criteria?

Reviewer #1: Yes



Reviewer #1: Yes

3. Have the authors made all data underlying the findings in their manuscript fully available (please refer to the Data Availability Statement at the start of the manuscript PDF file)?

Reviewer #1: Yes

4. Is the manuscript presented in an intelligible fashion and written in standard English?

Reviewer #1: Yes

5. Review Comments to the Author

Reviewer #1: Manuscript Title: "Challenges and enablers in measles vaccination implementation in Ethiopia: Insights from a qualitative study"

Journal: PLOS Global Public Health

Abstract

The findings should indicate the implementation challenges for service providers and caregivers. The way it is written does not show the challenges peculiar to caregivers or service providers.

Introduction

Line 44: It will be good to also know the number of deaths that have occurred in Ethiopia as a result of measles

Line 52: Cite the studies that have been conducted on barriers that prevent children from

53 getting the measles vaccine at the individual, community, and, to a lesser degree, service delivery levels. Also, let it be clear that the studies you’re referring to have been conducted in Ethiopia.

Methodology

Line 75-77: Provide a reference for the statement “Sodo and South Sodo districts were selected because of the high number of unvaccinated children in the area who needed to receive the measles vaccine”.

Line 88-89: What is the ideal saturation point of participant referred to in the statement “The study's sample size was determined based on the ideal saturation point of participants.”

Line 100: under data collection, did you develop the interview guide from other literature sources or you developed one from scratch? Kindly indicate and provide appropriate references if you developed from other literature sources.

Line 133: Please rephrase this sentence “Peer debriefing and audit trials are carried out as part of the second criteria, dependability, during data collection, translation, and transcription.” It is difficult to understand and also write in past tense. Also, can you explain clearly what transferability is?

Line 139: What certificate did you obtain from the University of Nicaragua? I think that sentence can be deleted.

Results

Line 169-170: Kindly explain the challenges explicitly before presenting the quotes. Since the study has different categories of respondents, it is necessary to let the reader know the challenges peculiar to the various respondents.

Line 286: Define any abbreviation first before you start using it. What is the meaning of AEFI?

Line 303: Are there no quotes to support the sub-theme “fear of contraindications”

The results are presented in a way that lacks coherence and structure, making it difficult for readers to follow the findings. Reorganize the results section to show a good linkage between the research objective and findings. Also, some findings are not sufficiently backed up with representative quotes.

Discussion

The discussion should be streamlined to avoid repetition and critically engage with the findings by comparing them with previous studies on measles vaccination barriers. Also, indicate the implications of the findings to policy and practice.

Comments

The study addresses an important but it requires significant improvements in the areas highlighted above. Also, the paper is full of grammatical errors and hence, the authors should thoroughly review the paper and improve on it.

6. PLOS authors have the option to publish the peer review history of their article (what does this mean? ). If published, this will include your full peer review and any attached files.

**Do you want your identity to be public for this peer review?** For information about this choice, including consent withdrawal, please see our Privacy Policy

Reviewer #1: No

---

## [Decision Letter · Decision Letter 1]

PGPH-D-24-02245R1

Challenges and enablers in measles vaccination implementation in Ethiopia: Insights from a qualitative study

Dear Dr. Wodajo,

Thank you for submitting your manuscript to PLOS Global Public Health. After careful consideration, we feel that it has merit but does not fully meet PLOS Global Public Health’s publication criteria as it currently stands. Therefore, we invite you to submit a revised version of the manuscript that addresses the points raised during the review process.

Reviewer 3 has just minor comments, along with Reviewer 4 (presented in an attached Word file). Respond to those and I would be able to quickly process this.

We look forward to receiving your revised manuscript.

Kind regards,

Abram L. Wagner, PhD, MPH

Academic Editor

Journal Requirements:

Additional Editor Comments (if provided):

Reviewers' comments:

Reviewer's Responses to Questions

**Comments to the Author**

Reviewer #2: All comments have been addressed

Reviewer #3: (No Response)

Reviewer #4: All comments have been addressed

publication criteria?

Reviewer #2: Yes

Reviewer #3: Yes

Reviewer #4: Yes

3. Has the statistical analysis been performed appropriately and rigorously?

Reviewer #2: Yes

Reviewer #3: No

Reviewer #4: Yes

4. Have the authors made all data underlying the findings in their manuscript fully available (please refer to the Data Availability Statement at the start of the manuscript PDF file)?

Reviewer #2: Yes

Reviewer #3: No

Reviewer #4: Yes

5. Is the manuscript presented in an intelligible fashion and written in standard English?

Reviewer #2: Yes

Reviewer #3: Yes

Reviewer #4: Yes

Reviewer #2: The needful correction has been done properly.

Reviewer #3: The study is interesting, relevant, educative, appropriate, and applicable in most PHC settings across the globe.

L1-3 Title: "Challenges and enablers in measles vaccination implementation in Ethiopia: Insights from a qualitative study" could be improved to reflect the aim of the study because the title speaks to the aim generally.

L13 Abstract is rather long and recommended between 250-350 words or not more than 2000 characters (no spaces).

Introduction has the first section repeating the words as the abstract.

The aim talks about the perceptions and experiences of implementing the measles vaccines challenges and enablers among healthcare workers and caregivers. However, perceptions could be part of the experiences.

L 16-18 Materials and Methods. L70 Case study design is multidisciplinary and seems appropriate in exploring lived experiences in that it triangulates data collection well but is laborious.

L74-79 Study setting: the purposively selection of sites is detailed enough.

L81-89: The 15 health workers, 16 mothers,, 15 exit interviews and six FGDs is rather too much. L90: Figure 1 should follow immediately for smooth reading. L43 rather say participants because respondents is reserved for quantitative studies. L92 please explain why such population is selected. How many refused to participant or dropped out and why?

L100 Data collection methods: State clearly who collected the data at each step. Duration data of interviews is 40-90 minutes. its too wide. What is the median duration? Any repeat interviews done and why?

What is meant by trustworthiness is accounted for? The robustness of the qualitative study depends of its trustworthiness and issues of credibility, transferability, dependability, conformability and triangulation. The application of all of them should be mentioned here.

L126 correct per "kebele".

L124-132 is a long paragraph with no meaningful sentences, separated by semi-colons and must be re-written.

L152 Results

L153 Socio-demographics of "respondents" - rather say participants.

Omit L154-160 and simply draw table 1. The one presented is too busy and may need separating health workers from caregivers or integrating it in a different way.

Data analysis: manually analyzed, how many independent coders were used and who coded what? The saturation points are rather obscured to a large extent. were recordings sent back to the participants to verify the correctness of the content?

L433 Discussion: From L435 the statements keep repeating the results. This section should focus on why such results were obtained and how they compare to other studies, rather than what is revealed.

L448-4458 Conclusions: this seems is repetition from the discussion. Rather use different wording.

Strengths and limitations or biases should include researcher bias, selection, confirmation, recall, Hawthorne effect, social desirability, interpretive etc. Also clearly state what was done to control it such as reflexivity, triangulation, member checking, thick description, peer debriefing etc.

L459: Abbreviations should in the text as they are encountered.

General comment:

-The study has huge potential for few publications, if the papers were to focus on specific aspects of the qualitative research methodology eg IDI, exit interviews or FDGs only and present separate findings of health workers from caregivers. The case study design complicates it because it becomes too larger to analyze manually.

Reviewer #4: All comments has been addressed

**Do you want your identity to be public for this peer review?** For information about this choice, including consent withdrawal, please see our Privacy Policy

Reviewer #2: **Yes: ** Dr. Md. Abdullah Yusuf

Reviewer #3: **Yes: ** John M. M. Musonda

Reviewer #4: No

---

## [Editor Report · Decision Letter 2]

Challenges and enablers in measles vaccination implementation in Ethiopia: Insights from a qualitative study

PGPH-D-24-02245R2

Dear Mr. Wodajo,

We are pleased to inform you that your manuscript 'Challenges and enablers in measles vaccination implementation in Ethiopia: Insights from a qualitative study' has been provisionally accepted for publication in PLOS Global Public Health.

Best regards,

Abram L. Wagner, PhD, MPH

Academic Editor

In the results you mention "Level 4" (health extension workers who have upgraded to Level 4) - can you explain or contextualize this?